# Analysis of Antibacterial and Antiviral Properties of ZnO and Cu Coatings Deposited by Magnetron Sputtering: Evaluation of Cell Viability and ROS Production

Viktors Vibornijs [1,†], Martins Zubkins [1,†], Edvards Strods [1], Zhanna Rudevica [2], Ksenija Korotkaja [2], Andrejs Ogurcovs [1], Karlis Kundzins [1], Juris Purans [1] and Anna Zajakina [2,*]

[1] Institute of Solid State Physics, University of Latvia, Kengaraga 8, LV-1063 Riga, Latvia; viktors.vibornijs@cfi.lu.lv (V.V.); martins.zubkins@cfi.lu.lv (M.Z.); edvards.strods@cfi.lu.lv (E.S.); andrejs.ogurcovs@cfi.lu.lv (A.O.); karlis.kundzins@cfi.lu.lv (K.K.); juris.purans@cfi.lu.lv (J.P.)

[2] Latvian Biomedical Research and Study Centre (LBMC), Ratsupites 1, LV-1067 Riga, Latvia; zhanna.rudevica@biomed.lu.lv (Z.R.); ksenija.korotkaja@biomed.lu.lv (K.K.)

\* Correspondence: anna@biomed.lu.lv

† These authors contributed equally to this work.

**Abstract:** The development and testing of antimicrobial coatings continues to be a crucial approach, considering the ongoing emergence of antibiotic-resistant bacteria and the rapid transmission of highly pathogenic viruses. In this study, three types of coatings—pure metallic copper (Cu), zinc oxide (ZnO), and a three-layer zinc oxide and copper mixed coating (ZnO/Cu/ZnO)—were deposited by magnetron sputtering on polyethylene terephthalate substrates to evaluate their antimicrobial potential using various microorganisms, including viruses. Gram-positive *Staphylococcus aureus* and Gram-negative *Escherichia coli* bacteria were used for the assessment of antibacterial properties. Antiviral testing was performed using MS2 bacteriophage and replication-deficient Semliki Forest virus, both representing single-stranded RNA-containing viruses. The samples' ability to cause reactive oxygen species formation was measured, and the effect on bacterial metabolic activity was evaluated. Cu-coated samples showed high inhibitory activity (>95%) against *E. coli* and *S. aureus* bacteria, as well as against tested viruses (SFV and MS2). The antibacterial and antiviral properties of ZnO/Cu/ZnO and ZnO coatings were not significant. Although ZnO/Cu/ZnO and ZnO caused inhibition of the metabolic activity of the bacteria, it was insufficient for complete bacteria eradication. Furthermore, significant reactive oxygen species (ROS) production was detected only for single Cu-coated samples, correlating with the strong bacteria-killing ability. We suppose that the ZnO layer exhibited a low release of Zn ions and prevented contact of the Cu layer with bacteria and viruses in the ZnO/Cu/ZnO coating. We conclude that current ZnO and Cu-ZnO-layered coatings do not possess antibacterial and antiviral activity.

**Keywords:** zinc oxide coatings; copper coatings; antiviral activity; antibacterial activity; reactive oxygen species; bacterial cell viability

## 1. Introduction

Surface coatings maintain an important role in determining a material's functionality and application range. The recent severe acute respiratory syndrome coronavirus 2 (SARS-CoV-2) pandemic, well-known as COVID-19, has also brought out the importance of antimicrobial surface coatings as one of the safety measures. Such anti-microbial agents could be effectively applied in healthcare and social facilities to prevent the spread of pathogens [1].

Magnetron sputtering technology provides the possibility to utilize a wide range of materials for creating thin-film coatings along with the flexibility to fine-tune preparation

parameters to obtain coatings with diverse properties. The technology is suitable for application on large-area surfaces and can be performed on various substrates, like polyethylene terephthalate (PET) sheets, different types of glass, metals, and even textiles [2,3]. Transparent coatings with anti-microbial, photochromic, electrochromic, electroconductive, and other effects can be obtained [4–7]. Surface modification with a magnetron-sputtered coating can also be utilized to enhance mechanical or chemical stability [4].

Zinc oxide (ZnO) is a versatile material with a wide range of applications, including antibacterial and antiviral properties. There are several studies on ZnO antimicrobial activity; however, most of them are related to the nanoparticulate form of ZnO [8–10]. The antimicrobial properties of ZnO thin films are less well investigated. The mechanism of ZnO antimicrobial activity can be attributed to (i) the direct interaction of nanocrystals with the cell membrane or virus envelope, (ii) the release of $Zn^{2+}$ ions, and (iii) the formation of reactive oxygen species (ROS) [8]. Furthermore, the photocatalytic activity of ZnO nanoparticles has been reported [11]. Although the primary anti-bacterial mechanism is unknown, it is supposed that $Zn^{2+}$ ions and ROS can damage bacterial cell membranes and DNA, leading to bacterial cell death.

The effectiveness of ZnO coatings can depend on factors such as composition, particle size, the structure of the ZnO material, and exposure to ultraviolet or visible light [12]. Alternatively, dopant additives like copper or silver can be employed to enhance the anti-microbial properties of ZnO [13]. Cu is one of the most studied materials, demonstrating a well-characterized antimicrobial property in coatings as it possesses a strong biocidal effect against bacteria and viruses. Even when applied in a pure metal state, Cu can cause the *E. coli* colony forming unit (CFU) log reduction rate to be up to $10^7$ [14]. It was supposed that the high antimicrobial efficiency occurs due to the material's chemical activity, ion production, and ROS formation [14]. However, Cu possesses several disadvantages as material degradation and corrosion occur.

It was shown previously that dielectric–metal–dielectric (DMD) coatings can serve as viable transparent electrodes in flexible optoelectronic devices [15,16], offering an alternative to the widely used but brittle indium tin oxide (ITO). ITO typically requires high production temperatures to achieve high conductivity and faces supply risks due to the natural scarcity of indium [17]. Recently, we have developed $WO_3/Cu/WO_3$ three-layer-structured transparent conductive coatings with high anti-microbial efficiency of up to $1 \times 10^5$ *E. coli* CFU reduction, where $WO_3$ layers are applied for transparency enhancement [4]. However, $WO_3/Cu/WO_3$ coatings are relatively chemically unstable due to Cu oxidation, which is reinforced by the amorphous structure of $WO_3$, and this instability influences the properties of the coating. Replacing $WO_3$ with ZnO could improve Cu protection because of the crystalline nature of ZnO. Additionally, Zn itself has higher oxidation potential than W.

In the present study, we used magnetron sputtering technology to deposit three-layer ZnO/Cu/ZnO coatings with the goal of achieving a more stable PET surface coating. The antibacterial properties were evaluated using Gram-negative and Gram-positive bacteria, and antiviral activity was assessed using enveloped and non-enveloped RNA viruses (a replication-deficient Semliki Forest virus or SFV and an MS2 bacteriophage). The SFV is an (+) ssRNA alphavirus of the *Togaviridae* family that has been previously employed as a model virus in antiviral studies [18,19] as it is structurally similar to some human disease-causing viruses, for example, SARS-CoV-2 [20]. A vector expressing the luciferase reporter gene (SFV/enhLuc) was used to enhance the sensitivity of virus quantification. The antimicrobial properties of ZnO/Cu/ZnO were compared with single ZnO and Cu coatings.

## 2. Materials and Methods

### 2.1. Deposition Technique and Characterization of Coatings

2.1.1. Magnetron Sputtering

Cu, ZnO, and ZnO/Cu/ZnO coatings on soda–lime glass, Si, and PET substrates were deposited by DC magnetron sputtering from Zn (PI-KEM Ltd., Tamworth, UK, purity 99.99%) and Cu (METALLIC FLEX GmbH, Habichtswald, Germany, purity 99.99%) targets. Two ION'X® planar balanced magnetrons (Thin Film Consulting, Grafenberg, Germany) with target dimensions of 200 mm × 100 mm × 9 mm were used. Both magnetrons were placed symmetrically against the substrate holder at a distance of 27 cm, as shown in Figure 1. The substrates were cleaned using an ultrasonic bath in acetone, followed by 2-isopropanol for 15 min each, then dried under a pressurised flow of $N_2$ gas. Thin-film deposition was performed using a G500M.1 vacuum PVD coater (Sidrabe Vacuum, Ltd., Riga, Latvia). Before the deposition process, the chamber ($\approx 0.1$ $m^3$) was purged down to a base pressure below $1.3 \times 10^{-3}$ Pa using a turbo-molecular pump backed with a rotary pump. The three-layer coatings of ZnO/Cu/ZnO were obtained in three consecutive deposition processes without breaking the vacuum as the chamber contained both of the utilised magnetron targets. ZnO films were deposited by sputtering the Zn target with a power of 300 W in an Ar (99.9999%) and $O_2$ (99.999%) atmosphere. The depositions were performed at a sputtering pressure of 1.3 Pa and a constant $O_2/Ar$ gas flow ratio of 2/3. The pumping speed was altered by a throttle valve to set the necessary sputtering pressure. Cu films were deposited by sputtering the Cu target with a power of 300 W at a total pressure of 0.7 Pa with only Ar gas present. The substrates were not heated intentionally during the deposition. The deposition parameters, as well as the thickness and optical and structural properties of the samples, are summarised in Table 1.

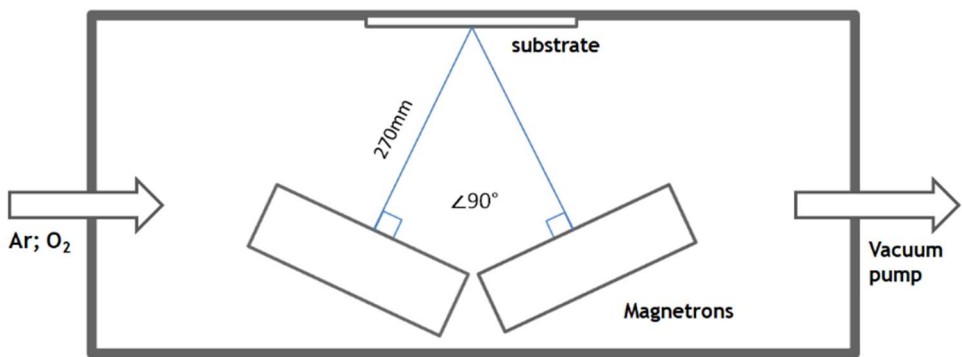

**Figure 1.** Schematic representation of magnetron locations and orientation.

**Table 1.** Summary of sample deposition parameters and physical properties. Deposition parameters: Ar and $O_2$ gas mass flow rates and sputtering pressure, ps. Physical properties: n&k values at 550 nm, surface roughness r, thickness obtained from spectroscopic ellipsometry (SE), mean squared error (MSE) of the SE models, average (400–700 nm) visible light transmittance $T_{400-700}$, and the crystalline structure (determined from X-ray diffraction (XRD)) of the samples.

| Sample | Deposition Parameters | | | Characterisation Results | | | | | | |
|---|---|---|---|---|---|---|---|---|---|---|
| | Ar (sccm) | $O_2$ (sccm) | ps (Pa) | SE | | | | | $T_{400-700}$ (%) | XRD Structure |
| | | | | MSE | $n_{550nm}$ | $k_{550nm}$ | r (nm) | d (nm) | | |
| Cu | 30 | 0 | 0.7 | 4.6 | 1.03 | 2.17 | - | 20 | 28 | Crystalline fcc (Fm-3m) |
| ZnO | 30 | 20 | 1.3 | 4.9 | 1.95 | 0.00 | 11 | 87 | 79 | Crystalline hcp ($P6_3mc$) |
| ZnO/Cu/ZnO | 30 | 20/0/20 | 1.3/0.7/1.3 | 4.6 | ZnO: 1.95 Cu: 1.27 | ZnO: 0.00 Cu: 1.55 | 13 | 80/20/98 | 29 | ZnO: Crystalline hcp ($P6_3mc$) Cu: Crystalline fcc (Fm-3m) |

### 2.1.2. X-ray Diffraction (XRD)

The crystallographic structure of the films was examined using X-ray diffraction (XRD), using a MiniFlex600 (RIGAKU, Tokyo, Japan) X-ray powder diffractometer with Bragg–Brentano $\theta$-$2\theta$ geometry and a 600 W Cu anode (Cu K$\alpha$ radiation, $\lambda$ = 1.5406 Å) X-ray tube.

### 2.1.3. Atomic Force Microscopy (AFM)

The surface texture of the deposited films was studied using a Park NX10 (Park Systems, Suwon, Republic of Korea) AFM in non-contact mode, using an ACTA type cantilever with a 10 nm tip radius.

### 2.1.4. Scanning Electron Microscopy (SEM)

The cross-section of the three-layer ZnO/Cu/ZnO coating was visualized using Helios 5 UX dual-beam SEM (Thermo Fisher Scientific, Waltham, MA, USA). The sample was prepared by scratching the coating off the substrate and imaging scratched pieces on the substrate, which are placed closely perpendicular to the substrate.

### 2.1.5. Spectroscopic Ellipsometry (SE)

Optical properties and film thicknesses were obtained by means of an RC2 spectroscopic ellipsometer (SE) (J.A. Woollam, Lincoln, MA, USA) in the spectral range from 5.9 to 0.7 eV. The main ellipsometric angles, $\Psi$ and $\Delta$, were measured at incident angles from 55 to 65° with a 5° step. The complex dielectric function dispersion curves (refractive index, n, and extinction coefficient, k) were modelled with the Cauchy and B-Spline dispersion equations. The surface roughness was modelled utilising effective medium approximation modelling performed with CompleteEASE 6.7 software.

### 2.1.6. UV-Vis-NIR Transmittance and Reflectance

The transmittance and reflectance of the coatings, in the range of 250 to 2500 nm, were determined by a spectrophotometer Cary 7000 (Agilent, Santa Clara, CA, USA). The sample was placed at an angle of 6° against the incident beam and the detector was placed at 180° behind the sample to measure transmittance and at 12° in front of the sample to measure specular reflectance.

### 2.2. Analysis of Antibacterial and Antiviral Properties

### 2.2.1. Bacterial Strains and Viruses

Antibacterial testing was performed using Gram-negative *Escherichia coli* (ATCC 25922) and Gram-positive *Staphylococcus aureus* (ATCC 25923) bacterial strains. Antiviral testing was performed using MS2 bacteriophage and recombinant replication-deficient Semliki Forest virus. The recombinant SFV was produced as previously described [21]. The MS2 was obtained from the bacteriophage collection of the Latvian Biomedical Research and Study Centre (a gift from A. Dislers).

### 2.2.2. Cultivation Media

Tryptic soy broth (TSB; Cat. No 610053; Liofilchem, Roseto degli Abruzzi, Italy): 17.0 g/L casein peptone (pancreatic); 2.5 g/L dipotassium hydrogen phosphate; 2.5 g/L glucose; 5.0 g/L sodium chloride; 3.0 g/L soya peptone (papain digest); pH = 7.3 ± 0.2 (25 °C).
Luria–Bertani broth (LB; Cat. No 610084; Liofilchem, Italy): 10.0 g/L tryptone; 5.0 g/L yeast extract; 5.0 g/L sodium chloride; pH 7.0 ± 0.2 (25 °C).
Soya casein digest lecithin polysorbate broth (SCDLP): 17.0 g/L casein peptone; 3.0 g/L soybean peptone; 2.5 g/L disodium hydrogen phosphate; 2.5 g/L glucose; 1 g/L lecithin (Cat. No 80007; Liofilchem, Italy); 7.0 g/L Tween 80 (Cat. No 412110; Liofilchem, Italy); pH = 7.3 ± 0.2 (25 °C).

### 2.2.3. Bacterial Tests

Testing was performed using a film covering method. The overnight cell culture was diluted with TSB medium to obtain $OD_{600} = 0.1$, which corresponds to approximately $1.5 \times 10^8$ CFU/mL. A 15 mm $\times$ 15 mm test piece was immersed in ethanol and completely dried. Each test piece was placed in a sterile 12-well plate, and 20 μL ($3.0 \times 10^6$ CFU) of cell suspension was placed dropwise on the testing surface and immediately covered by a sterile thin PET film 10 mm $\times$ 10 mm (0.04 mm thick). The 12-well plate, with test samples, was incubated for 2 h at room temperature. After incubation, the samples were resuspended in 980 μL SCDLP medium. To ensure careful cell detachment, the plate was placed on a shaker and rotated for 10 min at 220 rpm. Afterwards, 20 μL of the recovered cell suspension was transferred to the microplate with 180 μL of PBS for preparing the decimal dilutions. Diluted samples were plated on the agar plates by a microdilution droplet method. The number of colony-forming units (CFU) was counted after 24 h incubation at 37 °C. The conditions of test effectiveness were judged by preparing an additional sample with an untreated test piece (without coating) and test culture, which was immediately plated without incubation. The number CFU/mL was quantified for coated test samples and a PET control. The CFU log reduction rate and the reduction % were calculated according to the equations in Section 2.3.

### 2.2.4. MTT-Based Cell Viability Assay

To investigate the viability of bacterial cells, a 3-(4,5-dimethylthiazol-2-yl)-2,5-diphenyltetrazolium bromide (MTT; Cat. No 19265; USB, Affymetrix, CA, USA) assay was performed. Bacteria were grown in the TSB medium overnight (12–16 h). Then, the bacterial suspension in TSB ($OD_{600} = 0.1$ equal to $3.0 \times 10^6$ cells per 20 μL) was prepared. A 20 μL sample of the suspension was cultivated on the test surface covered with a film at RT. After 2 h, 980 μL of SCDLP media was added, and bacteria were resuspended and centrifuged at $3300 \times g$ rcf for 10 min. After removing the supernatant, the cell pellet was washed with 1 mL PBS. Cells were stained with 0.5 mg/mL MTT by resuspending the cell pellet in 100 μL of PBS and adding 10 μL of 5 mg/mL MTT solution to each tube. Pipetted samples were transferred to a 96-well plate. The plate was incubated for 1 h at 37 °C before 100 μL of solvent was added (5% SDS and 87.5% DMSO in ammonia buffer, pH = 10.00 (5.4 g $NH_4Cl$, 35 mL 10 M $NH_4OH$ in 100 mL)). The plate was incubated at 37 °C for 60 min before the optical density was measured with a Bio-Tek μQuant spectrophotometer (BioTek Instruments, Winooski, VT, USA) at 540 nm.

### 2.2.5. ROS Detection Assay

Bacteria were grown in the TSB medium overnight (12–16 h). The cells were washed three times with PBS before the bacterial suspension in PBS ($OD_{600} = 0.2$ equal to $6.0 \times 10^6$ cells per 20 μL) was prepared. Then, one volume of the bacterial suspension was mixed with one volume of 50 μM 2′,7′-dichlorodihydrofluorescein diacetate ($H_2DCFDA$; Cat. No 4091-99-0; Calbiochem) to allow the $H_2DCFDA$ compound to penetrate the cells. The suspension was incubated for 1 h at 37 °C, and then 20 μL ($3.0 \times 10^6$ cells) was cultivated on the test surface covered with film at room temperature. After 2 h, 200 μL of PBS was added, samples were pipetted, and 180 μL was transferred to a black, clear-bottomed 96-well plate. The plate was measured with a spectrophotometer Victor3V 1420-040 Multilabel HTS Counter (PerkinElmer, Shelton, CT, USA) at 485 nm (excitation) and 535 nm (emission) wavelengths.

### 2.2.6. Virus Tests

Semliki Forest Virus

The recombinant Semliki Forest virus (SFV), encoding the firefly luciferase gene (SFV/enhLuc), was produced as previously described [21]. Briefly, for the synthesis of infectious but replication-deficient virus particles, the BHK-21 cells (Baby Hamster Kidney cells; Cat. No. CCL-10™; ATCC/LGC Prochem, Boras, Sweden) were electroporated with

both recombinant viral RNA (SFV/enhLuc) and SFV helper RNA, providing a synthesis of SFV structural proteins. After 48 h incubation, the virus-containing medium was harvested.

The virus titres—infectious units per mL (iu/mL)—were quantified by infecting BHK-21 cells with serial dilutions of the virus, followed by immunostaining with rabbit polyclonal antibodies specific to the nsp1 subunit of SFV replicase (generously provided by A. Merits, Institute of Technology, University of Tartu, Estonia), as previously described [22].

The virucidal activity assessment was based on the JIS Z 2801 method [23] modified for virus tests. The fragment of the tested surface (6–9 mm × 6–9 mm PET) was placed into a 12-well plate, and a 10 μL droplet (in DMEM, Cat. No 41966-029; Gibco, Invitrogen) containing $7 \times 10^5$ iu of the SFV/enhLuc virus was placed in the centre of the PET. The virus droplet was covered by a piece of thin polypropylene plastic film and incubated for 30 min at RT. After incubation, the virus under the film was washed out with 600 μL of PBS (with $Mg^{2+}$ and $Ca^{2+}$).

In order to measure the virus infectious units, 200 μL of the recovered virus suspension was used for the infection of BHK-21 cells in a 24-well plate. BHK-21 cells were washed twice with PBS containing $Ca^{2+}$ and $Mg^{2+}$, and then recovered virus suspension was added to the cells. The cells were incubated at 37 °C for 1 h. Afterwards, the viral-particle-containing solution was aspirated and replaced with a Glasgow's MEM (Cat. No. 21710025; Gibco, Life Technologies, Thermo Fisher Scientific, Waltham, MA, USA) medium containing 1% foetal bovine serum (FBS; Cat. No. FBS-HI-12A; Capricorn Scientific, Ebsdorfergrund, Hessen, Germany), 10% tryptose phosphate broth solution (Cat. No. 18050039; Gibco, Life Technologies), 20 mM HEPES (Cat. No. 15630056; Gibco, Life Technologies), 2 mM L-glutamine (Cat. No. 25030024; Gibco, Life Technologies), 50 U/mL penicillin, and 50 mg/mL streptomycin (Cat. No. 15070063; Gibco, Life Technologies). The infected cells were incubated overnight at 37 °C and 5% $CO_2$. For the luciferase assay (Cat. No. E1501; Promega, Madison, WI, USA), the cells were lysed in 100 μL of cell culture lysis buffer (Cat. No. E1531; Promega) and centrifuged at $600 \times g$ rcf for 5 min, and 2 μL of the cell lysate was used immediately for the measurement of the luciferase enzymatic activity with the Victor3V 1420-040 Multilabel HTS Counter (PerkinElmer).

Virus titre was calculated by interpolating from the standard curve. Briefly, the correlation standard curve between relative luminescence (light) units (RLUs) and virus titre was quantified using standard SFV/enhLuc virus dilutions in a range of $5 \times 10^5$–$1 \times 10^3$ iu per infection. These standard dilutions of the virus samples were not incubated on surfaces. The virus titre standard curve was generated in each experiment, and the negative control signal (RLU of uninfected cells) was subtracted from all values.

The virus log reduction rate and the reduction % were calculated according to the equations in Section 2.3.

MS2 Bacteriophage

Bacteriophage propagation was performed by infection of the XL1-Blue *E. coli* strain in LB media supplemented with 0.2% chloroform. The MS2-bacteriophage-containing suspension was then centrifuged at $4000 \times g$ rcf for 10 min to remove *E. coli*. The suspension containing MS2 bacteriophage was then transferred to a new tube, again centrifuged at $4000 \times g$ rcf for 10 min, and filtered using 0.2-μm-pore-size filters to obtain a clarified MS2 bacteriophage suspension with a titre of $1 \times 10^{11}$ plaque-forming units (PFU).

The bacteriophage MS2 suspension was diluted in TSB to a concentration of $4 \times 10^7$ PFU/20 μL. Then, 20 μL of the suspension was applied to the test coating (1.5 cm × 1.5 cm) and covered with a PET film (1 cm × 1 cm). After 2 h of incubation at room temperature, the bacteriophage was washed from the sample. For the titre determination (PFU), a plaque assay was performed. Briefly, phage serial dilutions were prepared in LB medium. Indicator cells for infection (50 μL) were added (XL1-Blue, $OD_{600}$ = 1.0) to the Petri dish, followed by 50 μL of phage dilution. Four to five mL of 0.7% agar solution heated to 50 °C was poured into the mixture, and the plate was rotated to ensure

homogenous distribution. After waiting for the agar to polymerise, the plate was incubated at 37 °C for 24 h, and the plaques were counted:

$$plaque\ (pfu/mL) = plaque\ number \times dilution/volume$$

The bacteriophage log reduction rate and the reduction % were calculated according to the equations in Section 2.3.

2.2.7. Analysis of Soluble Fractions of the Coatings

To investigate whether soluble fractions derived from the coated surface cause the antibacterial effect in *E. coli*, a 20 μL drop of bacteria-free TSB medium was incubated on the surface for 2 h as previously described in Section 2.2.3. Then, the drop was resuspended in 180 μL TSB and 100 μL of this suspension was mixed with $1.5 \times 10^5$ *E. coli*. After 2 h incubation at RT, the cell suspension was mixed with 390 μL SCDLP, and microdilution assay was performed to quantify the survived cells as described above.

For ROS analysis, a 20 μL drop of bacteria-free PBS was incubated on the test surface for 2 h as described in Section 2.2.3. Then, the drop was resuspended in 200 μL PBS. A total of 180 μL of the suspension was mixed with 18 μL ($3.0 \times 10^6$ cells) $H_2DCFDA$-stained *E. coli*. The suspension was incubated for 2 h at RT, then the fluorescent signal was detected by fluorimeter as described in Section 2.2.5.

To investigate the antiviral effects of soluble fractions, virus-free DMEM medium was incubated on the test surface for 30 min, as for Section Semliki Forest Virus. The suspension was then mixed with $7 \times 10^5$ iu of SFV/enhLuc virus and immediately used for BHK-21 cell infection. The next day, the infectious capacity of viral particles was calculated using luciferase assay as described above.

*2.3. Statistical Analysis*

Statistical analysis was performed using GraphPad Prism 8.02 software. Data are expressed as mean ± standard error of the mean (SEM). An unpaired one-tailed Student's *t*-test was used to compare groups. Pearson's correlation analysis was performed to examine relationships. $p < 0.05$ was considered statistically significant (* $p < 0.05$; ** $p < 0.01$; *** $p < 0.001$; ns—non-significant). All tests were prepared in biological triplicate.

The log reduction was calculated according to the following equation:

$$log\ reduction = log_{10}(titre\ after\ incubation\ on\ PET) - log_{10}(titre\ after\ incubation\ on\ the\ surface).$$

The reduction % was calculated according to the following equation:

$$reduction\ (\%) = (1 - \frac{titre\ after\ incubation\ on\ the\ surface}{titre\ after\ incubation\ on\ PET}) \times 100\%$$

**3. Results**

*3.1. Structure and Optical Characterisation of Cu, ZnO, and ZnO/Cu/ZnO Coatings*

XRD patterns, recorded over a 2θ range of 5°–90° of the deposited Cu, ZnO, and ZnO/Cu/ZnO coatings are shown in Figure 2. The broad peak at 2θ ≈ 25° is due to the soda–lime glass substrate. A significant Bragg peak was observed at around 34° for the ZnO and ZnO/Cu/ZnO samples. This corresponds to the (002) plane of the wurtzite-type ZnO (w-ZnO) lattice with space group *P63mc* (Card No. 01-070-8072). It is well known that magnetron-sputtered ZnO films tend to grow highly textured with crystallites predominantly oriented with the c-axis perpendicular to the substrate surface [24]. The crystalline sizes in the [002] direction calculated by the Scherrer equation for the ZnO and ZnO/Cu/ZnO coatings are similar, 12 and 15 nm, respectively. In both cases, the lattice parameter, c, of 5.28 Å is larger than that of the w-ZnO crystal (c = 5.21 Å). The Cu and ZnO/Cu/ZnO samples show a low-intensity Bragg peak at 2θ = 43°, which originates from

the metallic Cu film in the *fcc* structure (Card No. 003–1018). The low intensity is due to the small thickness (20 nm) of the Cu film.

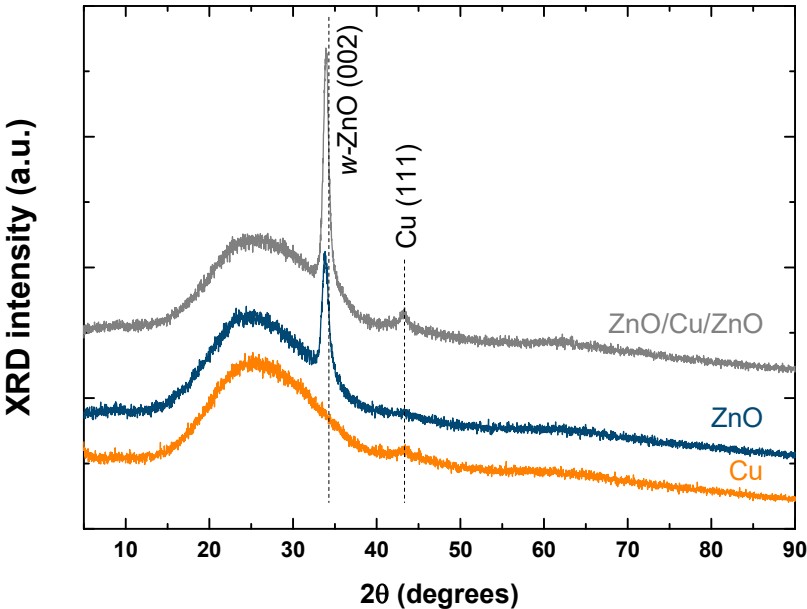

**Figure 2.** XRD patterns of the Cu, ZnO, and ZnO/Cu/ZnO coatings on soda–lime glass substrates.

AFM measurements of the deposited films revealed significant differences in surface morphology depending on the deposited material, as shown in Figure 3. The Cu film exhibited a relatively smooth, fine-featured surface with an average feature size of approximately 15 nm and a surface roughness of Ra = 0.552 nm. In contrast, the surface of the ZnO film comprised a conglomerate of larger features, closely packed, with an average size of roughly 50 nm and a surface roughness of Ra = 4.175 nm. Similarly, the ZnO/Cu/ZnO film presented a surface resembling that of the ZnO film, but with a noticeably smaller average feature size of about 30 nm and a surface roughness of Ra = 3.376 nm. Notably, the sample surfaces featured artefacts, likely associated with finer formations on the crystallite facets. In terms of antibacterial properties, a larger surface area is crucial, making materials with more extensive surface areas more effective.

Cross-sectional images of the ZnO/Cu/ZnO sample at two different magnifications obtained by SEM are shown in Figure 4. The images clearly depict the distinct three-layer structure of ZnO/Cu/ZnO. Additionally, the surface morphology is visible in Figure 4a from the scratched piece of the coating and in the background. This completely confirms the same morphological features obtained by AFM. A columnar growth with vertical voids perpendicular to the substrate surface is observed for the ZnO layers. The thickness of each layer measured from Figure 4b fits with the values (Table 1) extracted from SE modeling.

The results of the SE modelling are summarised in Table 1. Surface roughness should be introduced in the ZnO and ZnO/Cu/ZnO sample models to reduce the MSE values [25]. Values of 11 and 13 nm were obtained for the ZnO and ZnO/Cu/ZnO samples, respectively. In the case of pure Cu, surface roughness did not improve the model. The trend in roughness values obtained by SE coincides with AFM measurements (see Figure 3), indicating that the Cu film surface is significantly smoother than those of the ZnO and ZnO/Cu/ZnO coatings; however, SE models yield values approximately three times larger compared to AFM. The MSE values of all three models for the Cu, ZnO, and ZnO/Cu/ZnO samples were below 5. The models compared to the experimental SE data, as well as the n&k curves, are presented in Figure S1 of the Supplementary Material. The n&k values of the pure Cu film and the middle film in the ZnO/Cu/ZnO coating differ from the database values for pure Cu. This is due to the native oxidation of the Cu surface in air [26] or during reactive

deposition [27] of the upper ZnO film. The higher and lower values in our case agree well with the trend of optical coefficient variation between Cu and $Cu_2O/CuO$ phases [28].

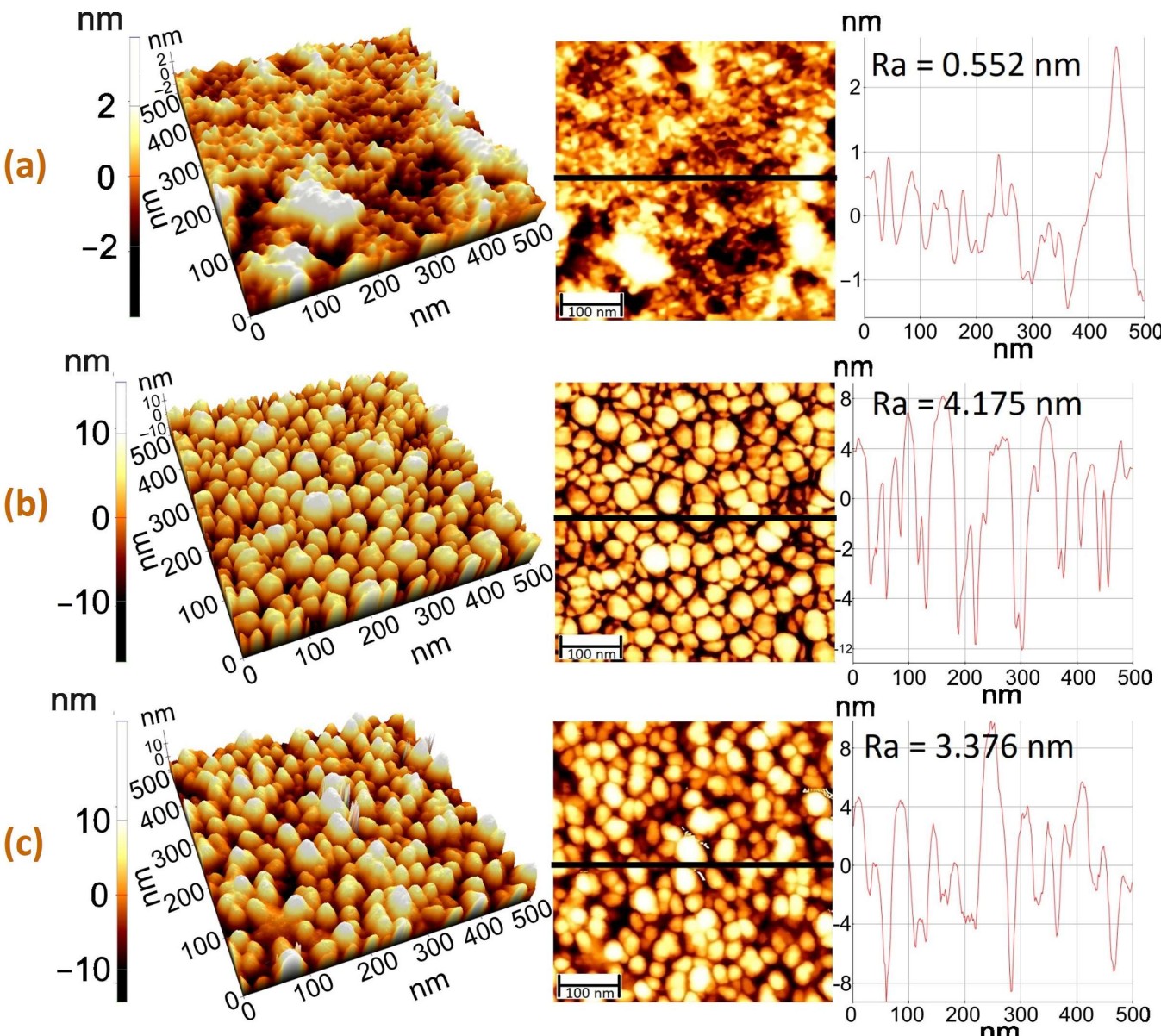

**Figure 3.** AFM surface morphology images and surface roughness scans of the (**a**) Cu, (**b**) ZnO, and (**c**) ZnO/Cu/ZnO coatings.

Figure 5 shows the specular transmittance and reflectance of the coatings deposited on soda–lime glass measured from 250 to 2500 nm. The pure ZnO film exhibits high transmittance in the visible light and near-infrared range, limited mainly by reflectance. The average reflectance in the visible light range, $R_{400–700}$, is 20%. Thus, the average transmittance in the visible range, $T_{400–700}$, is 79%. The sharp fall in transmittance below 400 nm is due to the onset of fundamental absorption of ZnO. Although SE predicts partial oxidation of the Cu film, it shows optical properties typical of thin metal films. The reflectance starts to increase strongly due to free electrons above the plasma edge at 565 nm, which is responsible for the natural reddish-brown colour. The 20-nm-thick Cu film exhibits a $T_{400–700}$ of 28% with a maximum of 33% at 575 nm. Due to suppressed reflection, the three-layer structure ZnO/Cu/ZnO shows higher transmittance in the near-infrared region than in the Cu sample.

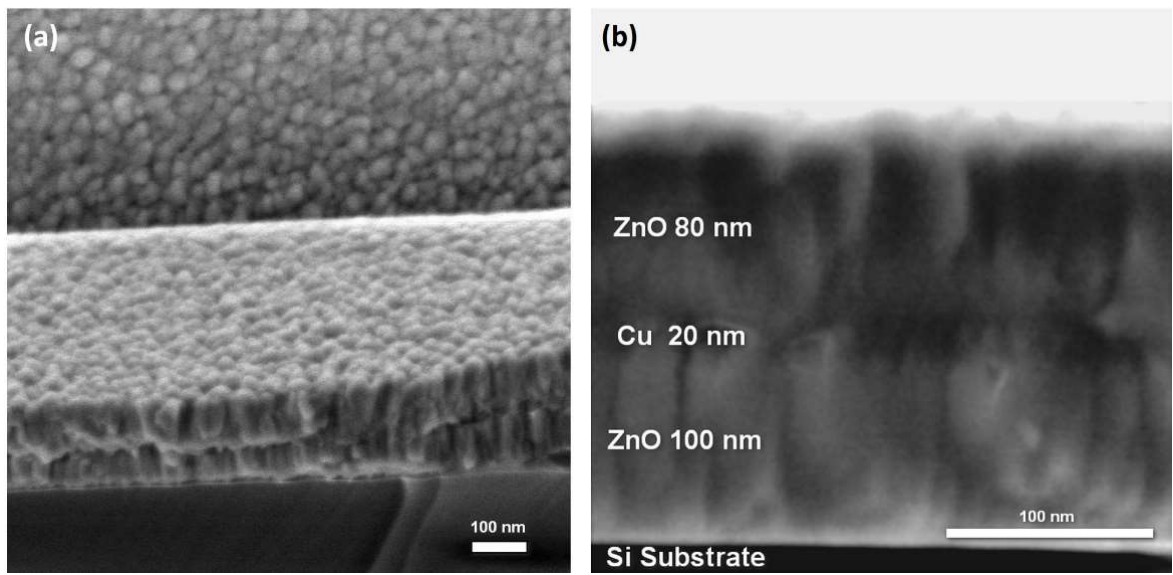

**Figure 4.** Cross-sectional SEM images at two different magnifications of the ZnO/Cu/ZnO coating. (**a**) at 100,000× magnification; (**b**) at 350,000× magnification.

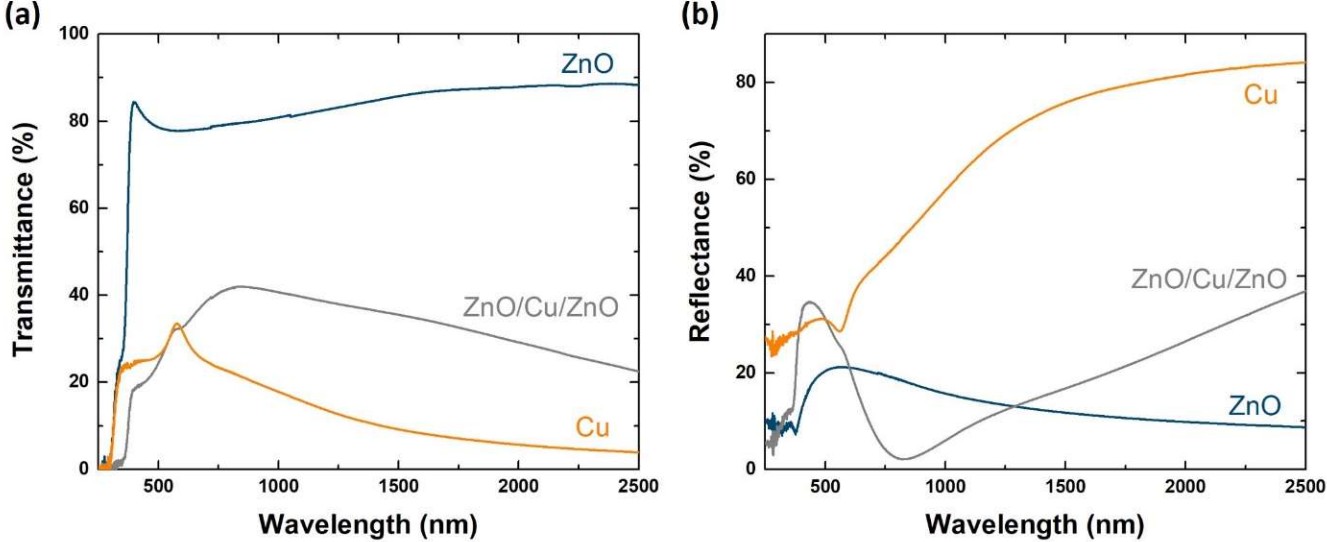

**Figure 5.** (**a**) Transmittance and (**b**) specular reflectance of the Cu, ZnO, and ZnO/Cu/ZnO coatings on soda–lime glass substrates in the range of 250–2500 nm.

### 3.2. Analysis of the Antimicrobial Activity of Cu and ZnO Coatings

The coated PET surfaces were tested using Gram-negative (*E. coli*) and Gram-positive (*S. aureus*) bacteria cultures as well as enveloped (SFV) and non-enveloped (MS2 bacteriophage) viruses. The overall coating application scheme is depicted in Figure 6. During the procedure, the suspension of the microorganism was applied to the coating and covered with a film. After the incubation at RT, the microorganisms were resuspended and used for quantification.

Bacterial growth was assessed by the ability to form colonies (CFU) after contact with the respective surface. The mechanism of cytotoxic effect was studied by analysis of metabolic activity and ROS formation, employing MTT and ROS assays, respectively. For the determination of antiviral properties, the titre of SFV virus was calculated by luciferase assay for SFV/enhLuc through the generation of a standard curve prepared from SFV/enhLuc virus dilutions with an appropriate number of infectious units (iu/mL). The quantification of MS2 bacteriophage was performed by a standard plaque assay.

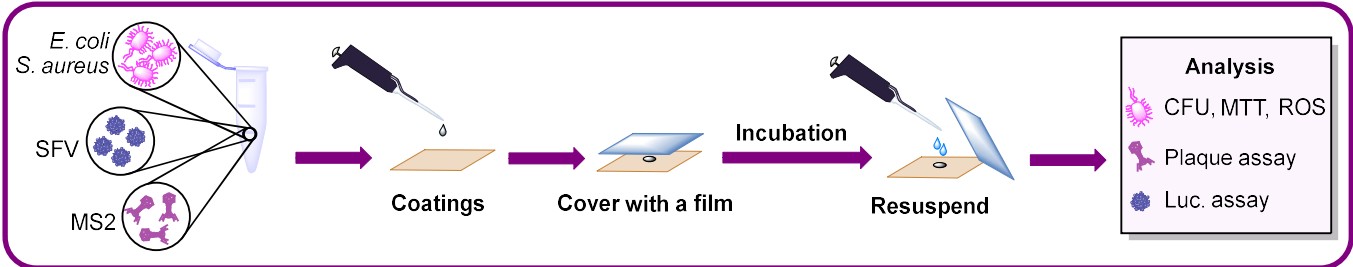

**Figure 6.** Schematic representation of the test workflow. The microorganism (bacteria or virus) suspension was placed on the test surface. The suspension was covered with a PET film and incubated at room temperature. After incubation, the microorganisms were collected and used for analysis. CFU, MTT, and ROS assays were performed to describe the antibacterial activity against *E. coli* and *S. aureus.* Plaque and luciferase assays were employed to quantify the titres of MS2 bacteriophage and Semliki Forest virus (SFV), respectively.

### 3.2.1. Analysis of the Antibacterial Activity of Cu and ZnO Coatings

Firstly, ZnO/Cu/ZnO, ZnO, and Cu coatings and PET as an uncoated control were tested with a Gram-negative *E. coli* bacterial culture. Briefly, $3.0 \times 10^6$ cells were incubated on coatings for 2 h. Bacteria were then collected and plated on the agar plates by a microdilution droplet method. The number of CFUs was counted after 24 h cell incubation at 37 °C. The CFU reduction rate was calculated relative to the PET control.

Cu coatings displayed an *E. coli* CFU log reduction rate of 1.3, which corresponded to a 95% *E. coli* reduction (Figure 7a). At the same time, the ZnO coating did not reduce the number of *E. coli* CFUs. However, the combined ZnO/Cu/ZnO coating sample exhibited a minor but significant reduction in CFUs (from $1.0 \times 10^6$ to $8.3 \times 10^5$, $p = 0.0215$) equal to 17% inhibition. Consequently, the incorporation of copper as a dopant in combination with ZnO enhanced the antibacterial properties of the coatings.

To obtain data about potentially incomplete *E. coli* inactivation caused by ZnO/Cu/ZnO, the cell viability of the bacteria was analysed by MTT assay (Figure 7b). Briefly, after incubation on tested coatings, bacteria were collected and stained with MTT. Bacterial samples were then incubated for 1 h at 37 °C, then the formed formazan crystals were dissolved and the signal was quantified. It was found that Cu reduced the metabolic activity of *E. coli* dehydrogenases and reductases by more than 98% (2% of viable cells). Moreover, both zinc-oxide-containing coatings reduced bacterial cell viability to 50%–60%. In the case of ZnO, *E. coli* viability was reduced to 56% compared with PET ($p = 0.0008$), while in the case of ZnO/Cu/ZnO it was 51% ($p = 0.0004$). Nevertheless, the coatings' effect on cell viability did not correlate significantly with bacteria CFU log reduction (Figure 7c, Pearson's correlation coefficient r = −0.8765, $R^2 = 0.7683$, $p = 0.1235$). This dependence reveals the lack of a complete cell inactivation effect by the ZnO and ZnO/Cu/ZnO coatings, indicating a cytostatic effect rather than a cell-killing effect.

To compare and potentially identify the precise mechanism by which the test samples affect bacterial cell viability, we measured ROS formation (Figure 7b). For ROS detection, *E. coli* were incubated on test sample surfaces in the presence of the ROS detection reagent 2′,7′-dichlorodihydrofluorescein diacetate ($H_2DCFDA$). Cu coating produced the highest ROS level among the tested samples—$3.4 \times 10^5$ relative fluorescence units (RFU). ROS formation was not detected in the case of pure ZnO. On the other hand, combined ZnO/Cu/ZnO coatings demonstrated minor but significant ROS induction compared to PET (from $4.5 \times 10^4$ to $9.7 \times 10^4$ RFU, $p = 0.0174$). ROS induction significantly correlated with the inhibition of bacterial number (Figure 7c, Pearson's correlation coefficient r = 0.9995, $R^2 = 0.9990$, $p = 0.0005$), highlighting the significance of ROS formation in achieving effective antibacterial activity.

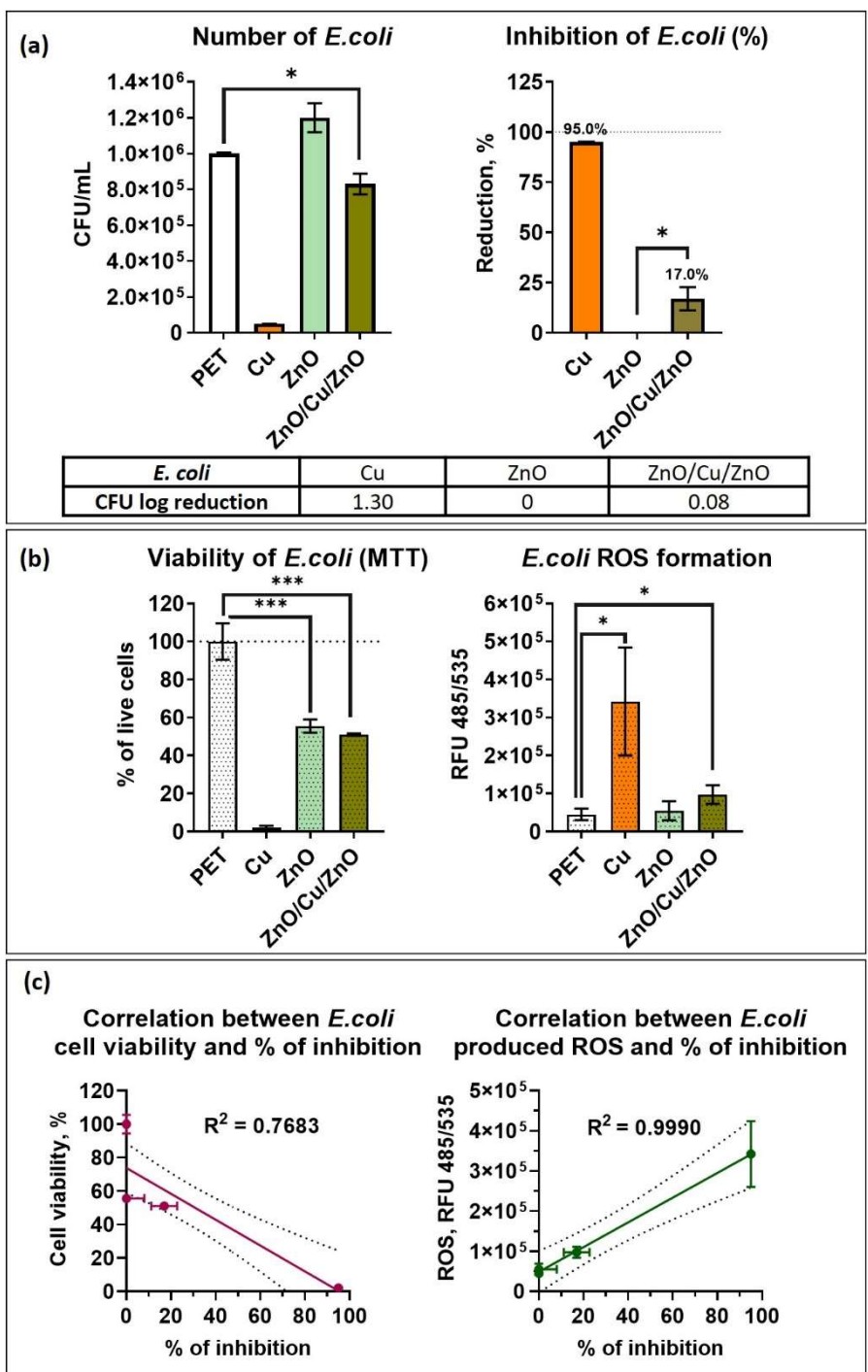

**Figure 7.** Antibacterial properties of the tested surfaces against *E. coli.* Briefly, $3.0 \times 10^6$ cells were incubated on coatings for 2 h, then the numbers of colony-forming units (CFU), viable cells (MTT), and produced reactive oxygen species (ROS) were determined. (**a**) Bacteria CFU log reduction and percent of inhibition. (**b**) Bacterial cell viability measured with MTT assay and ROS production. (**c**) Correlations between cell growth inhibition and cell viability (upper graph) and ROS (lower graph), respectively. Results are presented as mean ± standard error of the mean (SEM). PET—uncoated reference substrate, Cu—copper coating, ZnO—zinc oxide coating, ZnO/Cu/ZnO—combined zinc oxide and copper coating. *—$p < 0.05$, ***—$p < 0.001$.

Similar to the tests with *E. coli*, the ZnO/Cu/ZnO, ZnO, and Cu coatings and PET as an uncoated control were analysed with Gram-positive *S. aureus.* The Cu sample caused a CFU

log reduction of 3.7, which corresponded to a 99.98% bacterial cell reduction (Figure 8a). In contrast to the findings with *E. coli*, pure zinc oxide demonstrated a small reduction in CFUs (from $5.1 \times 10^5$ to $4.3 \times 10^5$, $p = 0.0251$) equal to 15.7% inhibition. However, in the case of the ZnO/Cu/ZnO combined sample, the reduction was not statistically significant for *S. aureus* CFU ($p > 0.05$).

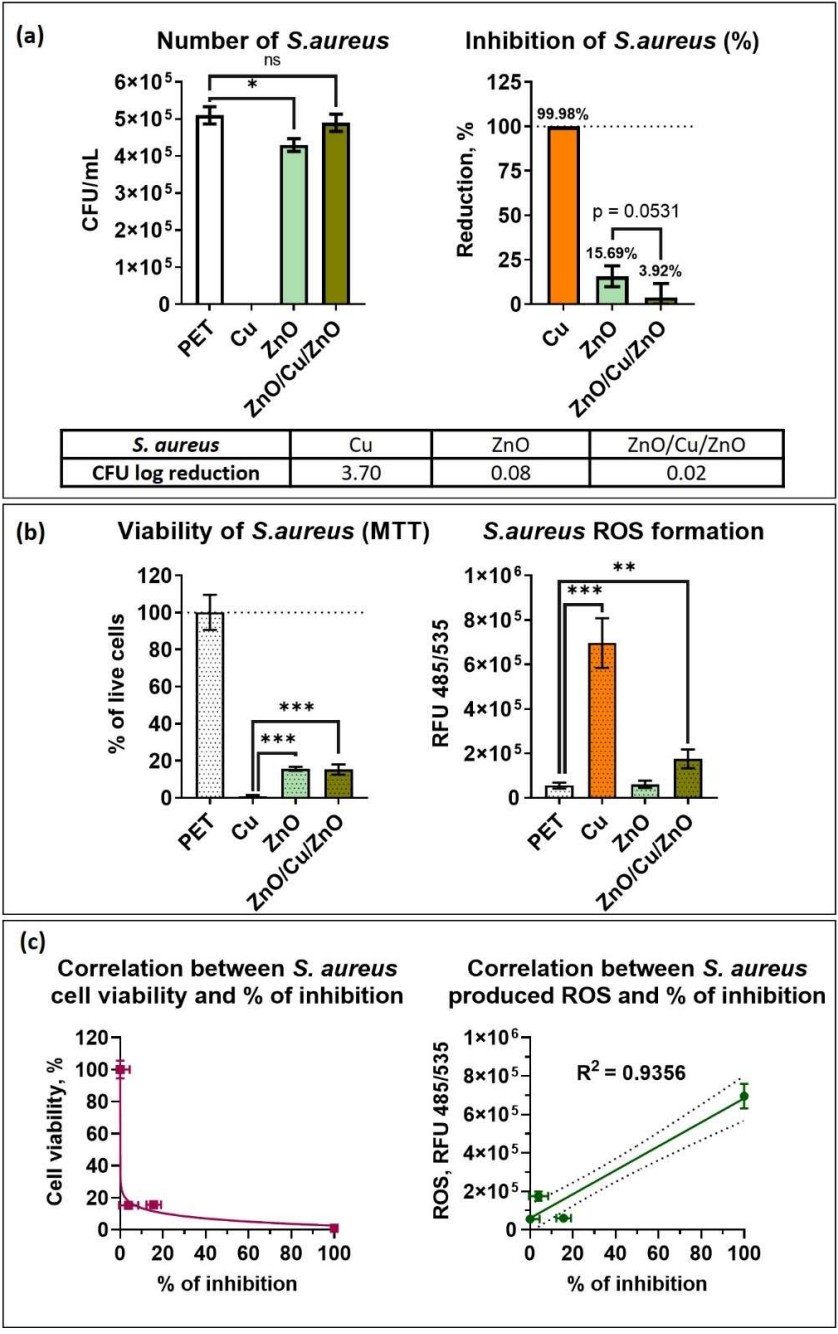

**Figure 8.** Antibacterial properties of the tested surfaces against *S. aureus.* Briefly, $3.0 \times 10^6$ cells were incubated on coatings for 2 h, then the numbers of colony-forming units (CFU), viable cells (MTT), and produced reactive oxygen species (ROS) were determined. (**a**) Bacteria CFU log reduction and percent of inhibition. (**b**) Bacterial cell viability measured with MTT assay and ROS production. (**c**) Correlations between cell growth inhibition and cell viability (upper graph) and ROS (lower graph), respectively. Results are presented as mean ± standard error of the mean (SEM). PET—uncoated reference substrate, Cu—copper coating, ZnO—zinc oxide coating, ZnO/Cu/ZnO—combined zinc oxide and copper coating. ns—non-significant, *—$p < 0.05$, **—$p < 0.01$, ***—$p < 0.001$.

*S. aureus* MTT assay revealed that all the surfaces reduce cell viability (Figure 8b). The copper layer reduced the cell viability by 99% (1% of viable cells), ZnO by 84.4% (15.6% of viable cells), and ZnO/Cu/ZnO by 84.7% (15.3% of viable cells). Comparable to *E. coli*, the impact of the coatings on cell viability did not exhibit a correlation with the reduction in bacterial CFUs (Figure 8c, Pearson's correlation coefficient r = −0.5532, $R^2$ = 0.3060, *p* = 0.4468). This dependence reveals the lack of a complete cell inactivation effect for ZnO and ZnO/Cu/ZnO coatings.

The results of the ROS assay were similar to those obtained with *E. coli* (Figure 8b): Cu coating led to the highest ROS formation level, equal to $7 \times 10^5$ RFU. ROS formation was smaller in the case of ZnO/Cu/ZnO ($1.8 \times 10^5$ RFU) and was not detected in the case of ZnO. Remarkably, in the case of *S. aureus*, ROS formation had a lower correlation with inhibitory effect (Figure 8c, Pearson's correlation coefficient r = 0.9673, $R^2$ = 0.9356, *p* = 0.0327) compared to *E. coli* (r = 0.9995, R2 = 0.9990, *p* = 0.0005, Figure 7c).

Notably, the antibacterial effect consistently exhibited greater efficacy in the case of *S. aureus* across all experiments. The Cu coating showed more effective inhibition of *S. aureus* resulting in a log reduction of 3.7 compared to 1.3 in the case of *E. coli*. Furthermore, the effect of ZnO-containing coatings on bacterial viability was 84%–85% in the case of *S. aureus* compared to 44%–49% in the case of *E. coli*. Moreover, *S. aureus* exhibited a higher ROS formation rate ($7 \times 10^5$ vs. $3.4 \times 10^5$ RFU for Cu and $1.8 \times 10^5$ vs. $9.7 \times 10^4$ RFU for ZnO/Cu/ZnO). The differential antibacterial effects observed can be attributed to differences in their structural characteristics. Gram-negative bacteria have an outer membrane that serves as an additional protective barrier. This outer membrane contains lipopolysaccharides, which can limit the penetration of antibacterial agents. On the other hand, Gram-positive bacteria lack this outer membrane, making them more susceptible to the action of antibacterial agents.

3.2.2. Analysis of the Antiviral Activity of Cu and ZnO Coatings

For the analysis of antiviral properties two types of model viruses were applied: SFV and MS2. First, SFV particles were employed as a model of an enveloped virus carrying an outer lipid envelope. SFV is an (+) ssRNA virus of the *Togaviridae* family. It possesses a close structural similarity to other human disease-causing viruses, for example, SARS-CoV-2 [20]. A virus vector expressing the luciferase reporter gene (SFV/enhLuc) was chosen to facilitate the accuracy and sensitivity of virus quantification. Moreover, the use of replication-deficient human virus in antiviral testing procedures is safe and much easier than using the wild-type infectious virus [18]. Briefly, after 30 min incubation on the tested coatings, the virus suspension was used for cell infection, then, 24 h post-infection, the infected cells were lysed and virus-mediated luciferase expression was quantified by introducing the luciferase substrate and measuring the emitted luminescence. The number of infectious units was determined using a standard curve generated from SFV/enhLuc virus dilutions.

The results for the anti-SFV effects are summarized in Figure 9 and Table 2. Virus was not detected in the case of the Cu surface, reflecting 100% virus inhibition and a >5.8 log reduction rate. Incubation on ZnO decreased the virus titre significantly from 6.2 to 4.1 iu/mL, *p* = 0.0076 (Figure 9a). This decrease is equal to a 0.2 log reduction and 34% inhibition (Figure 9b). However, virus reduction on the ZnO/Cu/ZnO coating was not statistically significant (*p* > 0.05).

Enveloped viruses are vulnerable to potential degradation of their lipid bilayer, while non-enveloped viruses typically demonstrate enhanced environmental stability, often thriving in extreme environments [29]. In this study, the non-enveloped (+) ssRNA bacteriophage MS2 (*Emesvirus zinderi*) was applied as a model of non-enveloped viruses.

MS2 bacteriophage was incubated on the tested coatings for 2 h and then used for infection of the host cells (*E. coli*). After 24 h, the MS2 titre was quantified by plaque assay (Figure 10). In the case of copper, the number of virus particles (plaque-forming units, pfu) decreased significantly (*p* = 0.0031), resulting in a >6 log reduction rate, similar to the

results obtained with SFV particles. No decrease in pfu, in comparison to the PET reference, was observed for the ZnO or ZnO/Cu/ZnO samples. Therefore, while zinc-containing samples exhibited a slight inhibitory effect on bacteria and enveloped viruses, they were not effective against non-enveloped viruses.

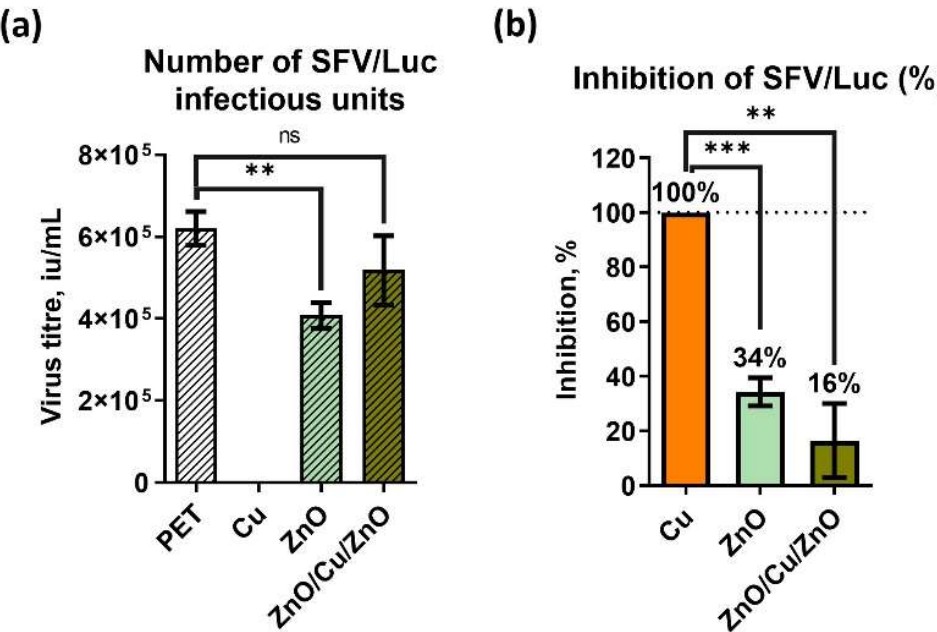

**Figure 9.** Antiviral properties of the tested surfaces against the Semliki Forest virus (SFV). To describe the decrease in virus infectivity, SFV expressing the luciferase gene was employed (SFV/Luc). Briefly, the virus was incubated on the test samples for 30 min, collected, and used to infect Baby Hamster Kidney cells (BHK-21). After 24 h, the number of infectious units (iu) was quantified using a luciferase assay. (**a**) Number of infectious units after incubation on test samples. (**b**) Inhibition (%) of SFV/Luc infectivity. Results are presented as mean $\pm$ standard error of the mean (SEM). PET—uncoated reference substrate, Cu—copper coating, ZnO—zinc oxide coating, ZnO/Cu/ZnO—combined zinc oxide and copper three-layer coating. ns—nonsignificant, **—$p < 0.01$, ***—$p < 0.001$.

**Table 2.** Antiviral properties of the tested surfaces against Semliki Forest virus (SFV).

|  | Virus Titer, iu/mL $\pm$ SEM | iu Log Reduction | Inhibition, % |
|---|---|---|---|
| PET | $(6.2 \pm 0.4) \times 10^5$ | Reference | Reference |
| Cu | $0.0 \pm 0.0$ | >5.8 | 100 |
| ZnO | $(4.1 \pm 0.3) \times 10^5$ | 0.2 | $34 \pm 5$ |
| ZnO/Cu/ZnO | $(5.2 \pm 0.8) \times 10^5$ | 0.1 | $16 \pm 14$ |

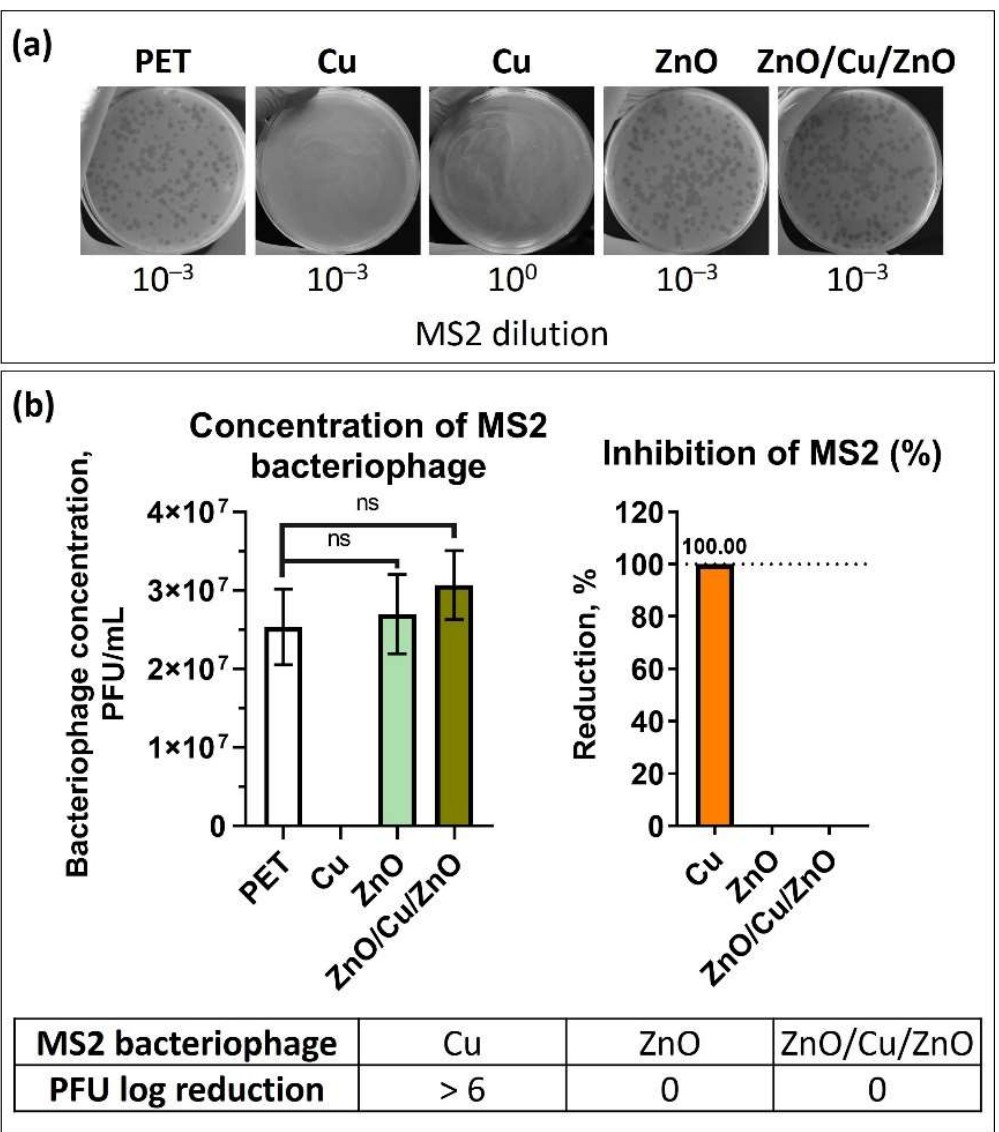

**Figure 10.** Inhibition of MS2 bacteriophage after 2 h contact with the test surfaces measured by *E. coli* culture infection and plaque-formation evaluation. (**a**) MS2 bacteriophage-formed plaques after MS2 incubation on the coatings. (**b**) Concentration of MS2 bacteriophage and inhibition (%) after incubation on test samples. Results are presented as mean ± standard error of the mean (SEM). PET—uncoated reference substrate, Cu—copper coating, ZnO—zinc oxide coating, ZnO/Cu/ZnO—combined zinc oxide and copper three-layer coating. ns—nonsignificant.

### 3.2.3. Analysis of the Soluble Fractions of Cu and ZnO Coatings

To test the biocidal activity of the soluble fractions, microorganism-free medium was incubated on the coatings, then its biocidal effect was evaluated using bacteria (*E. coli*) and a model virus (SFV).

First, we investigated the antibacterial effects of the soluble fractions of the coatings against *E. coli* (Figure 11a). Bacteria-free TSB medium was incubated on the surface for 2 h and then the suspension was mixed with *E. coli* and incubated for next 2 h at RT. After incubation, quantification of the surviving *E. coli* colonies was performed. Although some inhibitory effects of the Cu coating's soluble fraction were observed, this was not significant ($p = 0.1378$), demonstrating that indirect contact is not capable of causing a biocidal effect within a 2 h incubation with bacteria.

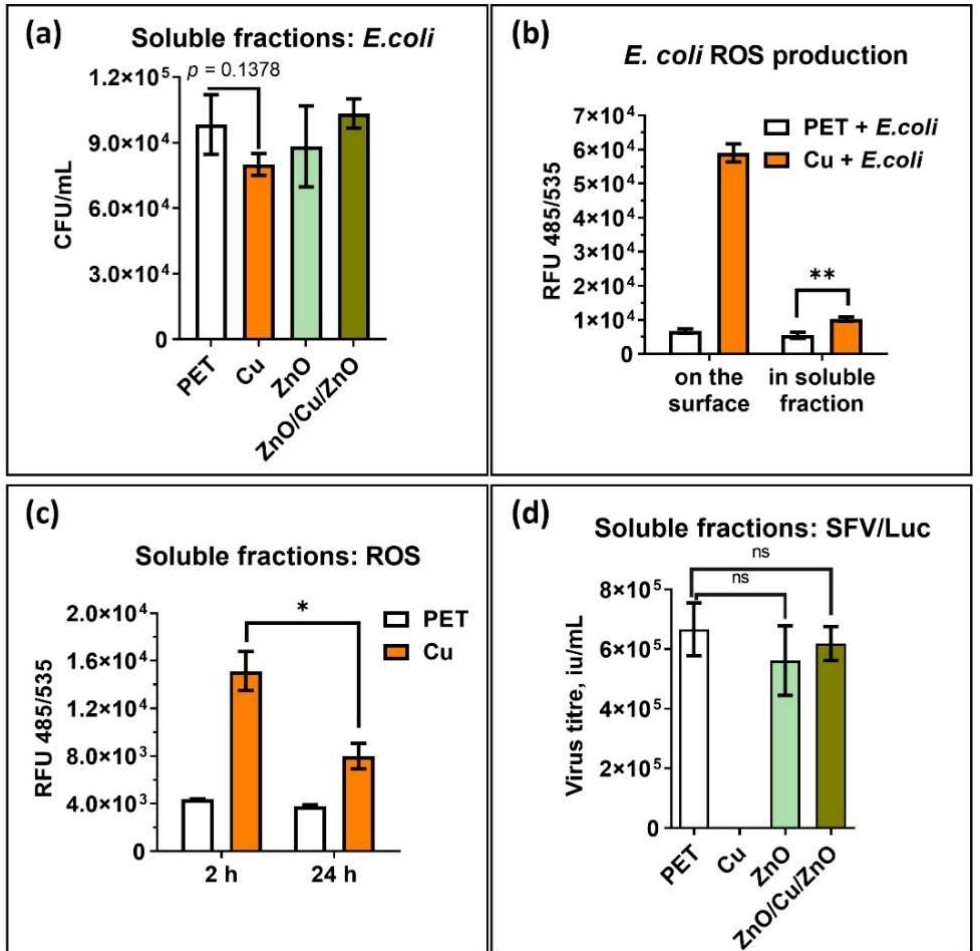

**Figure 11.** Antibacterial and antiviral activity of the coatings' soluble fractions. The medium without microorganisms was incubated on the test coatings for 2 h to achieve a soluble fraction. (**a**) Inhibitory effect of soluble fractions on *E. coli*. The soluble fraction was incubated with E. coli for 2 h, then the number of colonies was quantified. (**b**) ROS production after bacterial incubation on the Cu-coated surface and in the Cu soluble fraction. (**c**) ROS production without bacteria in the Cu soluble fractions preincubated for 2 h and preincubated for 24 h. (**d**) Number of infectious units after virus incubation with the soluble fractions. Results are presented as mean ± standard error of the mean (SEM). PET—uncoated reference substrate, Cu—copper coating, ZnO—zinc oxide coating, ZnO/Cu/ZnO—combined zinc oxide and copper three-layer coating. ns—nonsignificant, *—$p < 0.05$, **—$p < 0.01$.

As the Cu coating showed the best performance in the previous experiments, it was selected to test the production of ROS after bacterial incubation with the soluble fraction (in suspension). For this experiment, two strategies were compared: (i) *E. coli* were incubated with the medium preincubated for 2 h on the Cu coating (soluble fraction); (ii) *E. coli* were directly incubated on the surface of the Cu coating. After 2 h incubation, either with the Cu soluble fraction or on the Cu surface, the level of ROS was measured (Figure 11b). Indeed, the ROS produced in the suspension were significantly fewer than the ROS produced on the surface. Furthermore, we analysed the amount of ROS in the soluble fraction without *E. coli*. The soluble fractions achieved through medium incubation on the Cu surface for 2 and 24 h were subjected to ROS analysis. The soluble fractions were found to contain ROS; however, the ROS amount was lower after prolonged incubation (24 h), indicating the absence of a cumulative amount of ROS after 24 h (Figure 11c).

Second, it was tested whether the soluble fractions derived from the coated surfaces may cause the inhibition of SFV. Briefly, a medium without SFV virus particles was incu-

bated on the surfaces for 30 min as described in Figure 9. The suspension was then mixed with the virus particles and used for infection. No infective virus particles were detected in the case of Cu, showing the total inhibitory effect of the soluble fraction of the Cu coating (Figure 11d). However, in the case of the zinc surfaces, no statistically significant inhibition was detected, demonstrating the importance of virus–surface contact for effective inhibition. Remarkably, Cu-derived soluble factors are able to inhibit the enveloped virus but are not able to inhibit *E. coli*.

## 4. Discussion

In this study, we produced Cu, ZnO, and ZnO/Cu/ZnO coatings using the magnetron sputtering method, showcasing their crystalline nature (Figure 2). The corresponding samples are detailed in Table 1. The methods demonstrated in Figures 3 and 4 offer advantages over alternative thin-film deposition techniques [30,31], thanks to their ability to produce dense and smooth coatings with fine-featured surfaces and relatively low roughness. These characteristics make them particularly suitable for optical coatings, as the dense and smooth surfaces prevent optical properties from suffering diffuse light reflection [31,32]. Further insight into the potential modulation of optical properties across visible-to-infrared spectral regions is provided in Figure 5a,b, showcasing the impact of combining Cu and ZnO in a three-layer structure. This modulation could be tuned for various applications requiring antimicrobial surface properties. However, it is important to note that, in general, a higher surface area per unit volume, compared to flat surfaces in contact with microorganisms, leads to increased antimicrobial activity. The increased surface area provides more active sites for the antimicrobial process. Consequently, finding a balance between the optical properties and the antimicrobial activity of the magnetron-sputtered films is crucial.

ZnO has been the subject of extensive research in the fields of antibacterial and biomedical applications due to its potential advantages, including its chemical stability, optical transparency, mechanical durability, hydrophobicity, cost-effectiveness, low toxicity, high biocompatibility, and capacity to enable effective photocatalytic inactivation of bacteria [33,34]. ZnO coatings demonstrated antimicrobial activity against *E. coli* and *S. aureus* [35] and the incorporation of dopant metal (Cu) increased the antibacterial effects of ZnO coatings against *E. coli* [36]. In this study, we compared the antimicrobial properties of Cu, ZnO, and three-layer ZnO/Cu/ZnO coatings against model organisms representing Gram-positive and Gram-negative bacteria as well as enveloped and non-enveloped viruses. Tests with *E. coli* and *S. aureus* bacteria revealed that only Cu coating significantly decreased bacterial growth (>95%). Although the ZnO coatings did not produce a statistically significant decrease in CFUs compared to the Cu sample, the ZnO/Cu/ZnO coating showed up to a 17% inhibitory effect on *E. coli* (Figure 7), which is very low considering the high ability of bacteria to recover. We expected that three-layer ZnO/Cu/ZnO coatings possessing superior optical and electrical properties, as shown in a previous study [37], would demonstrate remarkable antibacterial effects. However, we observed only slight inhibitory effects. These findings are in agreement with previous studies describing the impact of ZnO coating thickness on its antibacterial effectiveness [38]. It is reported that thinner coatings (50 and 100 nm) possess very low antibacterial activity compared to the thickest ones (200–600 nm). There is a proportional rise in the active surface area and porous structure increasing contact and interaction with the bacteria. Our ZnO layer measured 80–90 nm in thickness (Figure 4), which can be insufficient for enhanced antimicrobial activity.

To investigate the nature of the antibacterial effects of the tested coatings, an MTT assay was employed. The assay was based on the enzymatic conversion of MTT by metabolically active cells into coloured formazan [39]. After contact with Cu coatings, the % of viable/metabolically active bacterial cells was very low (1%–2%) (Figures 7 and 8). This aligns well with the efficient inactivation of bacteria by Cu (CFU test). In the case of the ZnO and ZnO/Cu/ZnO coatings, the cell viability also significantly decreased. Moreover,

pure ZnO and combined ZnO/Cu/ZnO showed very similar results, reducing the viability of *E. coli* and *S. aureus*, comprising 51%–56% and 15%–16% of viable cells, respectively. This suggests that ZnO coatings may have adverse cytostatic effects on bacteria; however, these effects may not be sufficient to entirely kill the bacteria, resulting in a slight reduction in their growth rate. Practically, the MTT assay, as well as the ROS assay, can be used for fast screening of the antimicrobial activity of coatings, because the procedure is relatively less complex and not time-consuming compared with standard CFU quantification on agar plates.

It was reported previously that ZnO can be activated by exposure to ultraviolet (UV) or visible light [40,41]. Indeed, activation by light induced antibacterial properties of ZnO nanoparticles against *E. coli* and *S. aureus* [42]. However, Cu-doped ZnO coatings demonstrated an antibacterial effect against *E. coli* both in light (white light source) and in the dark [36]. The outcome can vary due to different amounts of dopant metal, incubation times, and the amounts of bacteria per test. Furthermore, UV treatment has a biocidal effect itself [43–45], which is difficult to adequately differentiate from the activity of the surface. We suppose that, in our study, the 80 nm ZnO layer protected the inner Cu layer from contact with the bacteria leading to inefficient inhibitory activity.

Although the molecular background of the bactericidal activity of antimicrobial coatings is not fully understood, the formation of reactive oxygen species (ROS) and related downstream metabolic inhibition is supposed to cause oxidative stress in microorganisms, involving membrane depolarisation, DNA and RNA oxidation, and protein deactivation or destruction [46,47]. Coating-derived ions can directly react with the organic molecules of cells or virus particles [48]. Various mechanisms of ion action have been described, e.g., effects such as ion binding to the active centres of enzymes and other proteins [49]. In this study, ROS formation in bacteria during incubation on coatings was tested. Cu-deposited samples induced substantial production of ROS upon incubation with both bacterial cultures (Figures 7 and 8). However, ROS were not detected in the case of ZnO. This can be explained by living bacteria's ability to degrade such compounds, for example with peroxidase group enzymes. Therefore, under such circumstances, it is not possible to exclude pure ZnO coating's ability to produce ROS, but it is completely obvious that the potential ROS production is not sufficient for complete bacterial inactivation. On the other hand, ZnO/Cu/ZnO showed a significant increase in ROS production, indicating a positive effect of ZnO in combination with Cu. Overall, ROS production significantly correlated with the inhibition of bacterial growth. This finding aligns with the results of previous studies utilizing ZnO nanoparticles to treat bacterial biofilms [50].

ZnO in the form of nanoparticles has demonstrated effective antiviral activity against various viruses, including hepatitis E and C, H1N1 influenza, herpes simplex virus type 1, and SARS-CoV-2 [51–54]. For antiviral testing, SFV, previously applied as a model virus in antiviral tests, was used [18,19]. SFV is an enveloped (+) ssRNA genome containing a human virus genetically modified for various biomedical applications [55]. In this study, we used replication-deficient SFV; therefore, the virus is capable of being used only for one round of cell infection, because new virus particles are not produced upon infection, representing a safe pathogen-related approach. The Cu sample caused complete SFV inactivation resulting in a >5.8 iu log reduction. Moreover, incubation on ZnO and ZnO/Cu/ZnO reduced virus titres by 34% and 16%, respectively (Figure 9). This inhibition cannot be considered an anti-viral effect compared to pure Cu.

Enveloped viruses (such as SFV, influenza, HBV, HCV, HIV, and human coronavirus) are at risk of outer lipid bilayer degradation, whereas non-enveloped viruses (such as rotavirus, norovirus, enterovirus, adenovirus, and rhinovirus) typically demonstrate remarkable environmental stability [29]. Consequently, we chose the non-enveloped (+) ss-RNA bacteriophage MS2 as a model of non-enveloped viruses. As expected, Cu-deposited samples showed 100% virus-killing properties in the tested system, whereas ZnO and ZnO/Cu/ZnO coatings did not reveal antiviral effects with MS2 RNA bacteriophage (Figure 10). Therefore, the antiviral properties of ZnO and ZnO/Cu/ZnO coatings are

not significant against enveloped or non-enveloped viruses. The biocidal performance of ZnO is widely attributed to the following mechanisms: (i) the release of $Zn^{2+}$ ions, (ii) the generation of ROS, and (iii) the direct attachment of microorganisms/viruses to ZnO nanoparticles. However, these mechanisms have primarily been studied in ZnO nanoparticle suspensions [56]. It is crucial to consider that coatings, in contrast to suspensions, have a notably smaller contact surface area. Therefore, transferring knowledge directly from suspension studies may not be applicable or straightforward to coatings. To determine whether the inhibition of infection was due to soluble ions/factors or direct surface contact, a microorganism-free medium was exposed to the surface to achieve a soluble fraction. *E. coli* and SFV incubation with soluble fractions of ZnO and ZnO/Cu/ZnO coatings did not reveal any inhibitory effects (Figure 11a,d). Interestingly, the Cu soluble fraction was highly inhibitory for SFV particles in contrast to *E. coli*, for which it demonstrated insignificant inhibition. It was shown previously that ZnO particles undergo partial dissolution in aqueous suspension, releasing $Zn^{2+}$ ions that play a significant role in the antimicrobial activity exhibited by ZnO [57,58]. Therefore, Zn ions could inhibit the growth of bacterial cultures [59–61]. However, no effect of soluble components was detected in our study against *E. coli*, which correlated with low ROS production. We suppose that the inability of the surface to release a sufficient amount of Zn and Cu ions in the case of ZnO/Cu/ZnO layered coatings is the reason for the absence of the antibacterial effect.

In a previous study, we developed $WO_3$/Cu/$WO_3$ three-layer-structured transparent conductive coatings that exhibited antimicrobial efficacy, achieving up to a 5-log reduction in *E. coli* CFUs, where $WO_3$ layers were utilized to enhance transparency [4]. However, these coatings, while effective, faced chemical instability, primarily due to Cu oxidation, compounded by the amorphous nature of $WO_3$. The substitution of $WO_3$ with ZnO could induce Cu protection due to the crystalline nature of ZnO. Furthermore, we suppose that the upper ZnO layer protects the inner Cu layer and blocks Cu ion release; therefore, the biocidal effect is weak.

In this study, we presented a comprehensive evaluation of the antimicrobial properties of ZnO and ZnO/Cu/ZnO coatings and compared them with pure Cu samples. Currently, there is no unified standard evaluation procedure for assessment of the antiviral and antibacterial properties of materials. Therefore, comparison of different published studies is difficult. Our experience indicates that the use of different bacterial strains and virus models affects the results significantly. Therefore, antimicrobial materials cannot be generalized or accepted to be efficient against all pathogens. Moreover, different factors affect antimicrobial efficiency, including the initial amount of bacteria or virus applied on the tested sample, incubation time, temperature, and quantification method. For example, in a previous study, a ZnO deposited sample demonstrated significant inhibition of *E. coli* and *S. aureus*, up to 100% [62]. However, the number of bacteria applied on the ZnO samples was $4 \times 10^3$ CFU/per sample, which is extremely low compared to those used in our study ($3.0 \times 10^6$ CFU/per sample), which made it possible to demonstrate up to a 6 log reduction rate. Here, we proposed to use the film covering method with testing parameters close to the international standards for measurement of antibacterial activity on plastics and other non-porous surfaces (ISO 22196:2011 [63]), which were adapted for the experimental procedure. On the other hand, in our study, the exposure time of the bacteria on the tested surface was shorter (2 h) and the incubation temperature was ambient, compared with the recommended ISO 22196:2011 standard (24 h, 35 °C). We applied Cu coating as a positive control, which demonstrated good anti-microbial performance within 2 h of incubation. Therefore, we concluded that 2 h is sufficient to demonstrate the anti-microbial efficiency of Cu/ZnO deposition. We suppose that extended incubation time (>2 h) may not accurately reflect the essential antimicrobial effects in a real environment, where rapid and efficient inactivation is required. Additionally, prolonged incubation could result in complete degradation of the coatings, rendering them less relevant for broader anti-microbial applications.

In summary, Cu coatings demonstrated superior antibacterial and antiviral activity. In the case of the three-layer coating, ZnO/Cu/ZnO, the upper ZnO layer prevents the inner Cu layer from reacting with microorganisms and viruses. Although the ZnO layer enhances the visible light transparency of the coatings, the lack of significant biocidal activity makes the current one-layer ZnO and three-layered ZnO/Cu/ZnO coating variants inappropriate for disinfection applications.

## 5. Conclusions

Cu showed strong inhibitory activity (>95%) against Gram-negative *E. coli* and Gram-positive *S. aureus* bacteria, as well as against the tested viruses—enveloped SFV and non-enveloped MS2 bacteriophage. The antibacterial and antiviral properties of ZnO/Cu/ZnO and ZnO coatings were not significant. Although ZnO/Cu/ZnO and ZnO caused inhibition of the metabolic activity of the bacteria, it was insufficient for killing the bacteria.

**Supplementary Materials:** The following supporting information can be downloaded at https://www.mdpi.com/article/10.3390/coatings14010014/s1, Figure S1: Main ellipsometric angles $\Psi$ and $\Delta$ (both experimental data and model fits) as a function of photon energy and incident angles and n&k curves for (a,b,c) ZnO, (d,e,f) Cu, and (g,h,i) ZnO/Cu/ZnO coatings.

**Author Contributions:** Conceptualization, M.Z. and A.Z.; methodology, Z.R., V.V. and K.K. (Ksenija Korotkaja), E.S., A.O. and K.K. (Karlis Kundzins); software, K.K. (Ksenija Korotkaja); validation V.V. and Z.R.; investigation V.V., Z.R. and K.K. (Ksenija Korotkaja); formal analysis, V.V., Z.R. and K.K. (Ksenija Korotkaja); resources, J.P. and A.Z.; data curation, M.Z., V.V. and A.Z.; writing—original draft preparation, V.V., K.K. (Ksenija Korotkaja) and M.Z.; writing—review and editing, A.Z. and M.Z.; visualization, K.K. (Ksenija Korotkaja), V.V. and E.S.; supervision, J.P., M.Z. and A.Z.; project administration, A.Z. and J.P.; funding acquisition, J.P. and A.Z. All authors have read and agreed to the published version of the manuscript.

**Funding:** This research was funded by the European Regional Development Fund (ERDF), Measure 1.1.1.1 "Support for applied research", Project No.: 1.1.1.1/21/A/050. AFM measurements by AO were supported by the State Education Development Agency, Project No. 1.1.1.2/16/I/001, Research Proposal No. 1.1.1.2/VIAA/4/20/590 "Portable diagnostic device based on a biosensor array of 2D material sensing elements".

**Institutional Review Board Statement:** Not applicable.

**Informed Consent Statement:** Not applicable.

**Data Availability Statement:** The data presented in this study are available on request from the first authors (VV, MZ) and corresponding author (AZ).

**Acknowledgments:** The authors thank Andris Dislers for providing the MS2 bacteriophage and for the establishment of a plaque assay methodology. The Institute of Solid-State Physics, University of Latvia, as a Center of Excellence, is thankful to the European Union's Horizon 2020 Framework Programme H2020-WIDESPREAD-01-2016-2017-TeamingPhase2 under grant agreement No. 739508, project CAMART[2] for provided equipment: Helios 5 UX dual-beam scanning electron microscope, RC2 spectroscopic ellipsometer, and Cary7000 spectrophotometer.

**Conflicts of Interest:** The authors declare no conflict of interest.

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
