# Peer review of "Analysis of Antibacterial and Antiviral Properties of ZnO and Cu Coatings Deposited by Magnetron Sputtering: Evaluation of Cell Viability and ROS Production"

_coatings, doi:10.3390/coatings14010014_

Round 1
Reviewer 1 Report
Comments and Suggestions for Authors
The manuscript from Vibornijs V. et. al. describes that the potential of Cu coating for anti-microbial activity is far superior to ZnO coating. This result is not consistent with some of the present literature, and the authors have not explained these differences convincingly. In addition to this, I have some major methodology questions as stated below so I will recommend major revisions before this manuscript is considered for publication.
· I have one major methodology question: Typically, for the ISO 22196 test, the contact (incubation) time is 24 hours at 37C, but the authors mention that they had a contact time of only 2 hours at room temp (presumable 25C?) (Page4 line 156) Can authors explain this major deviation from the ISO protocol?
· The same 2 hours and room temp protocol is repeated throughout the study, this makes me question the validity of all the results of all the microbial tests presented in this work from an ISO standpoint. If authors want to claim their study is per ISO 22196, additional experiments will be necessary.
· Have authors done a similar experiment as Page13 line 465 (effect of soluble factors) even for antibacterial activity of Cu surface against E. coli and S. aureus? These results make me wonder what the lifetime of antibacterial activity of the Cu surfaces will be. After what time will the Cu surface stop producing soluble ROS and Cu ions and exhaust its anti-microbial activity? Can authors comment on this aspect?
· Authors claim on page 16 line 598 that direct contact is necessary in the case of ZnO and refute the mechanism involving soluble Zn2+ mediated ROS generation. Have authors done a control to show that tiny (and comparable) levels of Zn2+ when directly added to bacterial culture do not cause any inhibition?
Author Response
Reviewer 1
Dear Reviewer,
Thank you for the detailed revision of our manuscript. We have revised it based on your comments and provided explanations of the changes that have been implemented.
Best regards
Sincerely yours
Anna Zajakina
Acting on behalf of all authors
The manuscript from Vibornijs V. et. al. describes that the potential of Cu coating for anti-microbial activity is far superior to ZnO coating. This result is not consistent with some of the present literature, and the authors have not explained these differences convincingly. In addition to this, I have some major methodology questions as stated below so I will recommend major revisions before this manuscript is considered for publication.
- I have one major methodology question: Typically, for the ISO 22196 test, the contact (incubation) time is 24 hours at 37C, but the authors mention that they had a contact time of only 2 hours at room temp (presumable 25C?) (Page4 line 156) Can authors explain this major deviation from the ISO protocol?
- The same 2 hours and room temp protocol is repeated throughout the study, this makes me question the validity of all the results of all the microbial tests presented in this work from an ISO standpoint. If authors want to claim their study is per ISO 22196, additional experiments will be necessary.
In our study, we have produced ZnO and layered ZnO/Cu/ZnO coatings to test their potential biocidal properties. Surprisingly, despite some evidence provided in the literature regarding the biocidal effect of ZnO, the anti-bacterial and anti-viral activity of these coatings was very low, comparing to pure Cu coating in 2h incubation. The mechanism of biocidal activity was suggested mostly in studies where the ZnO nanoparticles in suspension were used (https://www.ncbi.nlm.nih.gov/pmc/articles/PMC6223899/; https://www.ncbi.nlm.nih.gov/pmc/articles/PMC6057429/). In these studies, the release of Zn ions is much higher than what could be expected under the coated surface contact. Therefore, these data can not be directly attributed to magnetron sputtering-based coatings, which may differ in structure of ZnO upper layer. Therefore, the potential antimicrobial effects of all ZnO coatings remain to be speculative. Furthermore, the impact of ZnO coating thickness on its antibacterial effectiveness has been previously reported (https://www.sciencedirect.com/science/article/pii/S0169433214008368), demonstrating significant antibacterial effects only for layers >200nm. We have supplemented our study with cross-section SEM analysis of ZnO/Cu/ZnO coatings (please see Figure 4, page 11), which clearly shows the thickness of the ZnO layer up to 80 nm that could be insufficient for the appropriate amount of Zn ion release to exhibit significant antibacterial effects. Based on this observation and analysis of soluble factors (Figure 11, described below), we have supplemented the discussion section with respective explanations of inefficient antimicrobial effects of ZnO and ZnO/Cu/ZnO coatings (Please see page 23,24).
Regarding the methodology, namely the exposure time of bacteria on the tested surfaces, we have applied a shorter time (2 h, RT) compared to the recommended ISO 22196:2011 standard (24 h, 350 C) because we suppose that the extended incubation time (>2 h) and increased temperature may not accurately reflect the essential antimicrobial effects in a real environment, where rapid and efficient inactivation is required. Additionally, prolonged incubation could result in the complete degradation of the coatings, rendering them less relevant for broader anti-microbial applications. We used Cu coatings as a positive control, which demonstrated good anti-bacterial and antiviral effects after 2h, which was shown to be sufficient to evaluate these properties. In some other studies the shorter time (<24 h) was applied to demonstrate antibacterial efficiency:
https://pubs.acs.org/doi/10.1021/acsomega.7b00759
https://pubs.rsc.org/en/content/articlelanding/2014/TB/C4TB00196F
https://www.ncbi.nlm.nih.gov/pmc/articles/PMC9959704/
https://pubs.acs.org/doi/10.1021/acsami.1c15505 https://pubs.rsc.org/en/content/articlelanding/2016/TB/C5TB02312B
However, unfortunately, our ZnO and a three-layered ZnO/Cu/ZnO composition did not show any reliable anti-microbial effects, despite some literature data, making these coatings inappropriate compared to single Cu deposition. In that context, we believe that our results should be published to allow other scientists to analyze and consider these data for further development and optimization of ZnO-containing coatings.
We agree that referring to the ISO 22196:2011 standard is not adequate and introduces misapprehension in methodology. We have deleted the respective reference in Methodology section 2.2.3 (Page 5, line 182). However, to highlight the reliability of the 2h incubation time, the respective explanation was added in the discussion (pages 25,26, lines: 787-800).
- Have authors done a similar experiment as Page13 line 465 (effect of soluble factors) even for antibacterial activity of Cu surface against E. coli and S. aureus? These results make me wonder what the lifetime of antibacterial activity of the Cu surfaces will be. After what time will the Cu surface stop producing soluble ROS and Cu ions and exhaust its anti-microbial activity? Can authors comment on this aspect?
Authors claim on page 16 line 598 that direct contact is necessary in the case of ZnO and refute the mechanism involving soluble Zn2+ mediated ROS generation. Have authors done a control to show that tiny (and comparable) levels of Zn2+ when directly added to bacterial culture do not cause any inhibition?
A key contributing factor of the antimicrobial activity of metal-based surfaces has been attributed to the release of metal ions. Cu ions released from the surface were shown to possess antimicrobial activity (https://www.nature.com/articles/s41598-018-26391-8). It was also reported that ZnO nanoparticles undergo partial dissolution in aqueous suspension, releasing Zn2+ ions that play a significant role in the antimicrobial activity exhibited by ZnO
(https://www.sciencedirect.com/science/article/pii/S0927775714005172). Therefore, the released Cu and Zn ions potentially could inhibit the growth of bacterial cultures in our study.
We have performed additional experiments with soluble fractions achieved by medium preincubation (2h) with Cu, ZnO and ZnO/Cu/ZnO coatings (Figure 11). The preincubated medium (soluble fraction) was incubated with E.coli and SFV virus. No inhibitory effect of soluble components on E. coli was detected, probably due to the inability of the surface to release a sufficient amount of Cu and Zn ions for an antibacterial effect. In contrast, Cu soluble fraction but not ZnO fractions has totally inhibited the SFV virus, which appeared to be very sensitive for Cu ions. The respective description of activity of soluble fractions is presented in a new separate section 3.2.3 page 21, and in discussion section page 24,25, lines 737-762. These data support the idea that direct contact is important to exhibit antimicrobial activity. Furthermore, additional experiments were performed to assess ROS production in E.coli+Cu soluble fraction mixture, demonstrating ROS decrease compared to ROS production on the surface (the volumes, number of cells, the amount of the ROS reagent, and the incubation conditions were equal allowing the adequate comparison, please see the methodology section 2.2.7. Analysis of soluble fractions of the coatings, page 7 ). We cannot answer the question on when the Cu surface stops ROS production, however, we performed ROS analysis of Cu soluble fraction without bacteria after 2 h and 24 h (Figure 11d), which showed the lower ROS amount in 24h fraction, indicating the exhaustion of Cu surface after 24h. Therefore, these data demonstrate that the main antibacterial activity may happen before 24 h, and longer incubation will not bring additional benefits.
Regarding the tests with addition of Zn2+ ions as a control, it could be valuable in our experiments to estimate the amount of soluble Zn2+ ions required to reach efficient antimicrobial activity of coatings. Many studies showed the antimicrobial activity of Zn salts and other Zn containing compounds:
https://pubmed.ncbi.nlm.nih.gov/23509865/
https://pubmed.ncbi.nlm.nih.gov/11443104/
https://www.mdpi.com/1422-0067/22/10/5395
https://www.ncbi.nlm.nih.gov/pmc/articles/PMC243165/
However, in our opinion this control is not relevant to the current study, because it will not reflect the surface dynamic activity in the contact with microbes, furthermore, it is unclear what kind of compounds should be selected as a source of Zn ions, and how the release of Zn ions from the coatings can be controlled to compare with exogeneous Zn ions, because we expect it will be below the detection limits by methods available in our lab. On the other hand, the use of ZnO nanoparticles could be used as an additional control in future studies. Anyway, we are thankful for the idea of using exogenous metal ions as controls of our coatings.
Reviewer 2 Report
Comments and Suggestions for Authors
The authors have applied magnetron sputtering to fabricate a nanometer-thick, layered coating of ZnO/Cu/ZnO, aiming to demonstrate its antibacterial and antiviral effectiveness. However, the evidence presented does not sufficiently support the claim of its potential, thus I am unable to recommend the publication of this paper in Coating.
There is a particular concern regarding the comparatively lower antiviral and antibacterial efficiency of the ZnO/Cu coating in comparison with a Cu-coated surface. While the authors attempt to present an enhancement in antimicrobial properties of ZnO/Cu coating by referencing relative differences in parameters like bacteria/virus activity and inhibition rates between ZnO/Cu and ZnO coatings, the actual differences between the coatings are less than 30%. This margin does not substantiate a significant improvement in antimicrobial properties as claimed.
Additionally, while XRD and AFM data are provided to verify the successful deposition of Cu and ZnO layers, these techniques do not persuasively demonstrate that the materials have been coated in a layer-by-layer manner. It would strengthen the authors' case if they could include a cross-sectional scanning electron microscopy (SEM) image similar to what was presented in their prior study on WO3-Cu-WO3 coating. This would provide a more definitive illustration of the layered structure.
Author Response
Dear Reviewer,
Thank you for the revision of our manuscript. Please consider the changes that have been implemented to improve the manuscript. Please see the respective doc file with our reply.
Best regards
Sincerely yours
Anna Zajakina
Acting on behalf of all authors

Reviewer 3 Report
Comments and Suggestions for Authors
The article is very well written. Appropriate methods were used to obtain original coating compositions and to study their structure. Original studies of the antibacterial properties of the obtained coatings were carried out.
I have a few observations:
1. Captions (a), (b) and (c) in Figure 3 are in very large font.
2. Figures 6, 7, 8 and 9 are of very low quality. Fix the resolution.
The article can be published if the quality of Figures 3, 6, 7, 8 and 9 is improved.
Author Response
Dear Reviewer,
Thank you for the revision of our manuscript. We have revised it based on your comments. Please consider the improved version of our manuscript and reply provided to your comments.
Best regards
Sincerely yours
Anna Zajakina
Acting on behalf of all authors

Reviewer 4 Report
Comments and Suggestions for Authors
The manuscript “Analysis of Antibacterial and Antiviral Properties of ZnO and Cu Coatings Deposited by Magnetron Sputtering: Evaluation of Cell Viability and ROS Production” was written by Viktors Vibornijs and co-authored. The topic of new material layers with antimicrobial coatings is quite a curious and contemporary field of research. It has the potential to be applicable. The article has a good overall impression. It has an appropriate sectional structure for this type of research, which is presented in a concise, and clear manner. The introduction is sufficient background to the article. The discussion of research results is justified and supported by experiments. However, the organization and some figures could be presented in better quality. I have included all my comments in the list below:
1. Please expand in the introduction and explain why the authors conducted tests on the Semliki forest virus, SFV, (why this virus?).
2. The resolution of the figures should be higher to make the data easier to read. I also recommend reorganizing some of them. For example:
· Figure 3 – the surface roughness scans plots - the red line in the drawing should be thicker or darker.
· Figures 6 and 7 – please consider separating these graphs. It could have a positive impact on the resolution and readability of the figure. Perhaps moving part c) to the bottom of the figure would allow for a larger font. However, I suggest placing the table in Figure 8 c) as a separate table, not part of the draw.
· Figure 9 – Similarly in Figure 9, if the bar charts are moved above the figure or below the photos the font could be bigger and easier to read.
3. In Chapter 2. Materials and Methods, please complete information about the company, origin, and purity of all chemical reagents used.
4. The Supplementary Information requires author names and corrections in the title. Moreover, I recommended the separation and enlargement of graphs on Fig. S1.
Comments on the Quality of English Languagethe linguistic quality is correct. Minor editorial errors require correction, but they are minor and do not detract from the quality of the work.
Author Response

(The authors gave the same response as above.)

Round 2
Reviewer 1 Report
Comments and Suggestions for Authors
I think the authors have sufficiently addressed my concerns in the response letter and have made the required changes in the updated manuscript. I recommend accepting this manuscript without further revision.
Reviewer 2 Report
Comments and Suggestions for Authors
I appreciate the authors' responses to my questions. They have effectively demonstrated the antimicrobial properties of ZnO/Cu/ZnO and drawn logical conclusions in the revised manuscript with additional figures. This research stands to make a contribution to the design principles of inorganic antimicrobial coatings. Therefore, I recommend its publication in Coating.